# Visible-Light Photocatalyst to Remove Indoor Ozone under Ambient Condition

**Jia Quan Su** [1,2] , **Yi-Chun Chang** [1] **and Jeffrey C. S. Wu** [1,*]

1   Department of Chemical Engineering, National Taiwan University, Taipei 10617, Taiwan; jiaquansu.alan@gmail.com (J.Q.S.); r07524056@ntu.edu.tw (Y.-C.C.)
2   Department of Atmospheric Sciences, National Taiwan University, Taipei 10617, Taiwan
*   Correspondence: cswu@ntu.edu.tw

**Abstract:** Ozone is a kind of hazardous gas in indoor areas and needs to be removed in order to protect the human respiratory system. Previous methods include physical adsorption, thermal treatment, electromagnetic radiation removal, catalysis and photocatalysis. However, they all have limited effects. This research introduced a novel milestone to remove indoor ozone by utilizing visible light photocatalysis technique under ambient condition. The modified sol–gel method was applied to prepare photocatalysts, strontium titanate ($SrTiO_3$) and rhodium-doped strontium titanate ($SrTiO_3$:Rh). In addition, the $SrTiO_3$:Rh was further immersed in N3 dye to improve its photocatalytic performance. Batch system and continuous-flow system were used to quantify the removal rate of ozone and to measure the conversions of ozone, respectively. The results showed that $SrTiO_3$:Rh possessed a higher ozone removal rate under a visible light condition compared with a commercial P25 $TiO_2$ catalyst. In addition, $SrTiO_3$:Rh based catalysts can also successfully perform visible light ozone photodecomposition in the continuous ozone flow system. Note that current ozone converters in aircraft utilize thermal-catalysts, which can only be operated at high temperature. This research reveals a promising catalysts and photo process, which can possibly replace the current aircraft ozone converters with visible-light driven converters, and boast higher performance under ambient condition.

**Keywords:** photocatalysis; ozone decomposition; strontium titanate; visible-light converter

## 1. Introduction

Ozone has been a controversial gas in the atmosphere. While the role of ozone in the stratosphere is to prevent our Earth from excess ultraviolet-light radiation, ozone can also act as a kind of greenhouse gas and photochemical smog pollutant. Ozone in the troposphere is mainly produced by the oxidation of carbon monoxide and hydrocarbons. However, ozone concentration in open areas can seldom exceed the regulated amount of ozone concentration except for highly polluted areas [1–8].

According to regulations from Environmental Protection Agency (EPA), Taiwan, the average ozone concentration should not exceed 0.12 ppm within 1-hour exposure. Yet, in indoor areas, especially in some factories, ozone can easily accumulate to a higher concentration, which is extremely harmful to human respiratory systems [9–12]. For example, in semiconductor factories, silicon wafers usually go through the etching process with UV light sources in most cases. Therefore, ozone is possibly produced by dissociating the bond between oxygen atoms under radiation. Furthermore, the environment is not ventilated well in most printing factories, and ozone will eventually accumulate to a high concentration and jeopardize workers' health. In addition, aircraft cabins may also face this problem. Airplanes mainly fly at a high altitude so that a high concentration ozone may enter the cabin with bleed air, which frequently result in an odorous smell [13]. Several companies have realized this issue and start developing ozone converters to remove indoor ozone. Thus, an ozone converter is potentially a high profit product, which may be

commercialized and sold to every airline company. However, the current patent on ozone converters for aircrafts mainly used manganese dioxide as the catalyst. This kind of catalyst suffers from a significant decrease in activity under high relative humidity [14,15]. As a result, our goal is to explore a novel material, which can work under ambient conditions to remove indoor ozone, and can potentially replace the current commercial catalyst.

Previous research have utilized several different methods to remove indoor ozone besides manganese dioxide. One way is to use activated carbon as the adsorption substrate for ozone removal. However, only a limited amount of ozone can be adsorbed on this material, which also needs to be replaced regularly [16–18]. Another way to remove indoor ozone is by thermal treatment, but this method results in high energy consumption and is not efficient [19–21]. Further investigation also shows the possibility of using electromagnetic radiation to remove ozone. Nevertheless, radiation of short wavelength might activate the bond between oxygen molecules and produce even more ozone [22–24]. Thus, scientists have come up with a promising way to remove indoor ozone, that is, catalytic decomposition [25–33].

However, among all the materials used in catalytic decomposition, multiple disadvantages exist. The first known thermal catalyst used for ozone decomposition is manganese dioxide, but the active sites on the surface of the catalyst are occupied by water vapor and cause the catalyst to lose its catalytic ability [14,15,30,34–36]. Some groups tried doping noble metals or synthesized composite materials to support manganese dioxide, but they can only show a high catalytic removal rate at a high initial concentration of ozone under suitable temperature and humidity conditions [14,26,37–40]. Another recent research utilized ZnO based material for ozone catalytic decomposition [41]. Although the material shows a moderate ozone degradation rate, the reaction is operated under high initial ozone concentration, i.e., 20 ppm, and the fabrication process requires delicate defect engineering [41]. Therefore, some researchers turn their target to investigate the potential of using photocatalysis to remove indoor ozone. The most common material, which has been proved to possess high photocatalytic activity is titanium dioxide [42–50]. Although titanium dioxide does have a good ozone photocatalytic decomposition rate, it only absorbs UV irradiation, so it can not be widely utilized in factories or aircraft cabins since these areas are mainly under visible light condition. Consequently, it is important to find a material, which has good photocatalytic activity under visible light and also able to remove ozone at a diluted concentration. Gong et al. [46] utilized nitrogen-doped $TiO_2$, hydrogenated $TiO_2$ and commercial P25 for visible-light photoreaction. Additionally, they have conducted the experiment with a low power white LED and with a shorter irradiation time to show the total ozone removal efficiency (including adsorption and photocatalysis). However, it is still important to distinguish the ozone removal between adsorption and real photocatalysis. Furthermore, the system should be operated for prolonged time interval to rule out the ozone removal ability by adsorption in $TiO_2$-based material.

Photocatalysts possess semiconductor properties, when light irradiates on photocatalysts, holes and electrons are generated. In addition, the photogenerated electrons can react with ozone and reduce it to ozone radical ($O_3^-$) as shown in Figure 1. Ozone radical is a highly active reactant and can be easily decomposed into oxygen and oxygen radical under light irradiation [44].

The photocatalysts used in this research were a well-known perovskite structure material, strontium titanate ($SrTiO_3$) and rhodium-doped strontium titanate ($SrTiO_3$:Rh). Additionally, commercial catalyst P25 is used to compare the photocatalytic activity. According to Dvoranova et al. [51] and Konta et al. [52], the bandgap of strontium titanate is 3.2 eV, and it is possible to lower the bandgap to 2.3 eV by doping rhodium into the structure. The lowered band gap of rhodium-doped strontium titanate enables visible light absorption, paving the way for the ultimate goal of this research, which is to demonstrate the ozone photodecomposition effect under visible light condition. A previous study has shown that modifying the Rh dopant concentration could influence the light absorption efficiency, the density of electron traps, and the internal quantum yield [53]. In addition, when

$SrTiO_3$:Rh is immersed in N3 dye, the catalyst surface might have a certain amount of dye physically adsorbed on it. Additionally, this process can further increase the visible-light absorption of photocatalysts.

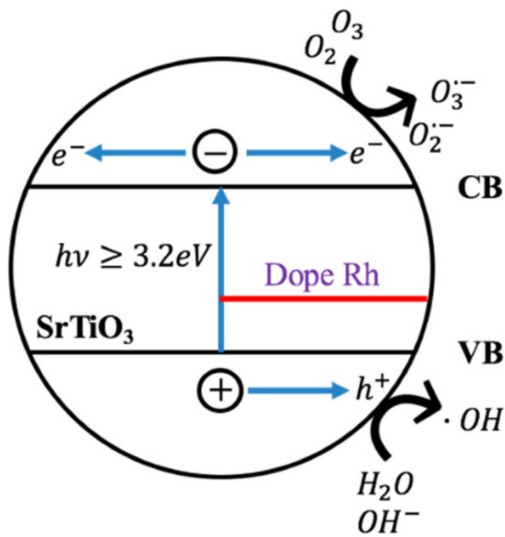

**Figure 1.** Mechanism of ozone photodecomposition.

The outcome of this research showed that our catalyst can effectively remove ozone at a diluted concentration to meet EPA regulations, and we achieved photocatalytic decomposition of ozone under the visible light condition. Furthermore, we developed a modified sol–gel method to synthesize strontium titanate based catalysts and showed the potential to commercialize these materials since the production process is straightforward in the solution phase.

## 2. Results and Discussion

### 2.1. Batch System

#### 2.1.1. Background Test

A background test is conducted without a catalyst or light irradiation as a base for comparison among the results of batch system. From Figure S3, we can see that the ozone concentration undergoes a natural decay after the ozone generator is switched off due to the high reactivity of ozone.

#### 2.1.2. Batch System—UV Irradiation

For the following results, with the same catalyst, three repeated tests will be shown on the same figure and the initial concentration are manipulated to be identical by controlling the ozone production period. Catalyst amount in each test is 0.1 g.

According to Figures S4–S6, the red lines are all a blank test, which indicates the experiment without adding a catalyst but applied with a light source. The other three lines show the photocatalytic decomposition of ozone under UV (365 nm) irradiation by using the same catalyst for three times. The results show that by removing ozone from the same initial concentration, the less time requires, the faster the ozone removal rate is. In Figures S4 and S5, several decay lines were above the red blank decay, these were suspected to be experimental errors. Furthermore, it was observed that P25 could provide the highest ozone removal rate under UV light. However, it is difficult to separate the ozone decomposition contribution between adsorption and photocatalysis in the batch system. Therefore, the continuous flow system was also applied to show the contribution of both factors respectively.

### 2.1.3. Batch System—Visible Light Irradiation

Generally, P25 and $SrTiO_3$ mainly absorb light and show photocatalytic ability in the UV region while $SrTiO_3$:Rh can absorb visible light to perform photocatalytic ability. Therefore, three kinds of different catalysts were tested in both light sources in order to compare the results. In this section, visible light photocatalytic decomposition results for batch system are shown.

From the visible light irradiation results (Figures S7–S9), we can observe that every test with the catalyst will result in a higher ozone removal rate than the blank test, which shows that with catalyst addition, it is truly able to remove ozone at a higher rate. However, there are several aspects, which still have to be investigated. Since P25 and $SrTiO_3$ only absorb a little amount of visible light, maybe the ozone removal is mostly contributed by catalyst adsorption. Therefore, the operation on the continuous flow system will start the photocatalytic decomposition reaction after the catalyst reaches a saturated adsorption amount.

### 2.1.4. Ozone Removal Rate in the Batch System

The ozone removal rate for P25, $SrTiO_3$ and $SrTiO_3$:Rh under UV (365 nm) irradiation and visible light irradiation (with AM 1.5G filter) are listed systematically in Table 1.

**Table 1.** The ozone removal rate for three catalysts in a batch system under two different light sources.

| Light Source | Catalyst (Type-Cycle) | Ozone Removal Rate ($\frac{\mu mol}{gcat \cdot min}$) |
|---|---|---|
| Visible light | $SrTiO_3$:Rh-1 | $8.1 \times 10^{-3}$ |
| | $SrTiO_3$:Rh 2 | $2 \times 10^{-3}$ |
| | $SrTiO_3$:Rh-3 | $1.77 \times 10^{-2}$ |
| | $SrTiO_3$-1 | $6.6 \times 10^{-3}$ |
| | $SrTiO_3$-2 | $4.9 \times 10^{-3}$ |
| | $SrTiO_3$-3 | $5.6 \times 10^{-3}$ |
| | P25-1 | $6.6 \times 10^{-3}$ |
| | P25-2 | $1.0 \times 10^{-2}$ |
| | P25-3 | $8.1 \times 10^{-3}$ |
| UV light | $SrTiO_3$:Rh-1 | $1.5 \times 10^{-3}$ |
| | $SrTiO_3$:Rh-2 | $1.0 \times 10^{-4}$ |
| | $SrTiO_3$:Rh-3 | $0$ |
| | $SrTiO_3$-1 | $5.1 \times 10^{-3}$ |
| | $SrTiO_3$-2 | $5.1 \times 10^{-3}$ |
| | $SrTiO_3$-3 | $1.5 \times 10^{-3}$ |
| | P25-1 | $1.05 \times 10^{-2}$ |
| | P25-2 | $1.17 \times 10^{-2}$ |
| | P25-3 | $1.05 \times 10^{-2}$ |

As we average the ozone removal rate for the same catalyst, the results can be easier as shown in Figure 2. From Figure 2, in the UV light region, P25 possesses the high ozone removal rate 0.011 (µmol/gcat/min) while $SrTiO_3$:Rh only has a much slower ozone removal rate. In Figures S4 and S5, the difference of the blank test curves and the sample test curves are narrow, which indicates that $SrTiO_3$ and $SrTiO_3$:Rh are efficient under UV irradiation comparing to the ozone natural decay. In contrast, in the visible light region, $SrTiO_3$:Rh possesses a higher ozone removal rate than commercial catalyst P25, and the rate was nearly $9.3 \times 10^{-3}$ (µmol/gcat/min). Although P25 can perform quite well in both light sources, $SrTiO_3$:Rh still has the potential to be the favorable material to remove indoor ozone since most of the indoor illumination comes from visible light. In addition, $SrTiO_3$ does have a moderate amount of ozone removal rate in both light sources. However, in the batch system, it is difficult to distinguish the contribution between catalyst adsorption and photocatalytic decomposition. Random errors may exist including, whether the gas in the reactor is well mixed, whether the catalyst is uniformly distributed for light illumination,

whether the ambient light from the environment is blocked well. Therefore, by utilizing continuous flow system, saturation of catalyst adsorption can be reached before applying light irradiation to observe photocatalytic performance.

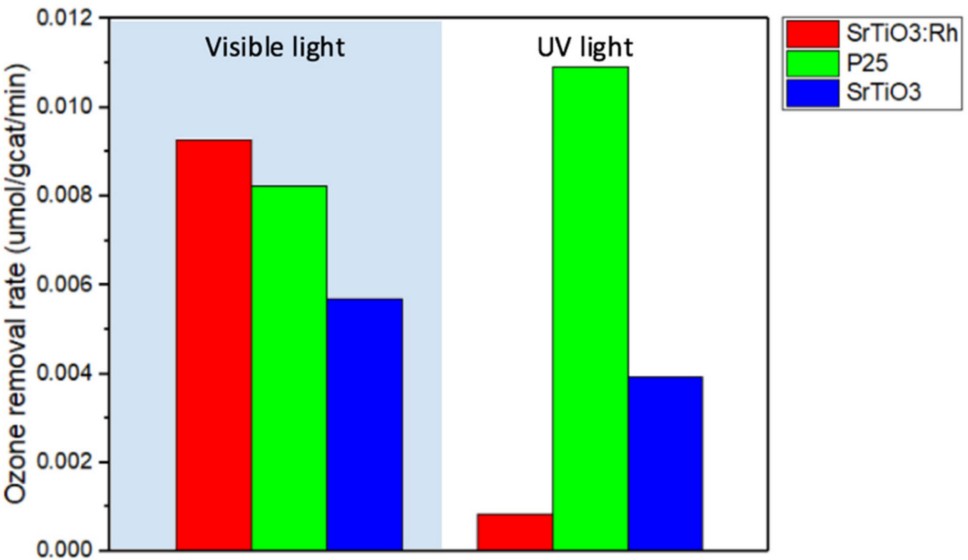

**Figure 2.** The averaged ozone removal rate in the batch system.

*2.2. Continuous Flow System*

2.2.1. Pre-Experiment Setup

In the continuous flow system, the system requires over two hours to reach the steady state first. During this process, the catalyst is already placed inside the reactor for adsorption. When the adsorption/desorption has reached the equilibrium state, the ozone concentration maintains at a steady-state value, and the light is then turned on.

2.2.2. Continuous Flow System—UV irradiation

For the test in continuous flow system under UV irradiation, the space velocity is adjusted by a rotameter to be 168 L $g^{-1}$ $h^{-1}$. The results for three different catalysts (P25, $SrTiO_3$ and $SrTiO_3$:Rh) are shown in Figures S10–S12. Additionally, the blue background color in the figures indicates the time interval in which the light is on.

The photocatalytic decomposition of ozone is performed for three continuous cycles as shown in Figures S10 and S11. The concentration has a dramatic decrease when the light is turned on and remains at a relatively lower ozone concentration for ten minutes, then the light is turned off in order to observe how the concentration rises back. Additionally, this process is repeatedly carried out for three times throughout the test for each material. As shown in Figure S11, the horizontal line for the light off condition has a gradually decreasing trend because the adsorption/desorption rate of ozone on $SrTiO_3$ is slower than P25. So, that the concentration of the light off condition will gradually decrease as the fixed parameter is the time length between each cycle. Additionally, by comparing the difference of ozone concentration between light-on and light-off conditions among three catalysts, the results show that P25 had the largest photocatalytic ability to remove ozone while $SrTiO_3$:Rh had nearly no photocatalytic ability to remove ozone under UV irradiation. Tests under a higher flow rate, 7250 L $g^{-1}$ $h^{-1}$, were also conducted, and the results are shown in Table 2. Due to a higher flow rate, the effect of P25 in the ozone removal was the only one significant enough to be detected.

**Table 2.** Ozone removal rate in a flow system under two different light sources (SV = 7250 L $g^{-1}$ $h^{-1}$ UV light: 365 nm pen light, OmniCure; visible light: Halogen lamp, 50 W, broad band, OSRAM, Germany).

| Light Source | Catalyst | Ozone Conversion (%) |
|:---:|:---:|:---:|
| Visible light | N3-SrTiO$_3$:Rh | 7.3 |
| | SrTiO$_3$:Rh | 5.6 |
| | P25 | 3.2 |
| UV light | SrTiO$_3$:Rh | 0.0 |
| | SrTiO$_3$ | 0.0 |
| | P25 | 6.3 |

### 2.2.3. Continuous Flow System—Visible Light Irradiation

For the test in the continuous flow system under visible light irradiation, the space velocity was settled at a higher flow rate, which was 7250 L $g^{-1}$ $h^{-1}$. The results for three different catalysts (P25, SrTiO$_3$:Rh and N3 dye sensitized SrTiO$_3$:Rh) will be shown in Figures 3–5. The blue background color in the figures also indicates the time interval in which the light is on.

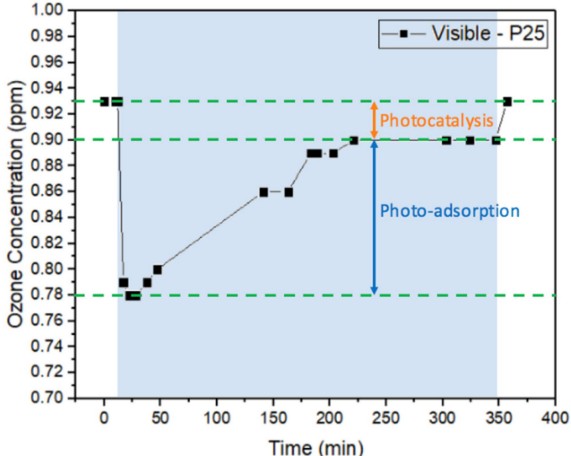

**Figure 3.** Photocatalytic decomposition of ozone in the continuous flow system by P25 under the halogen lamp (OSRAM, 50 W, broad band, with 400–500 nm bandpass filter).

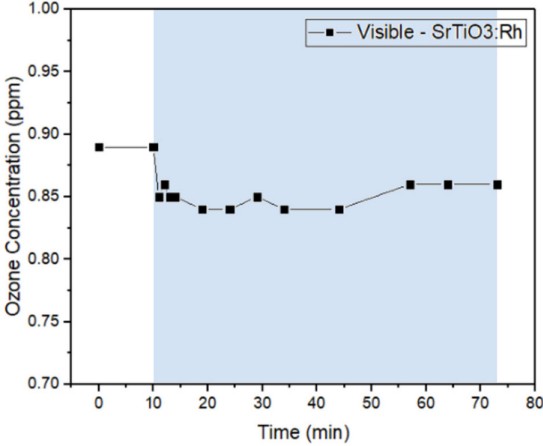

**Figure 4.** Photocatalytic decomposition of ozone in the continuous flow system by SrTiO$_3$:Rh under the halogen lamp (OSRAM, 50 W, broad band, with 400–500 nm bandpass filter).

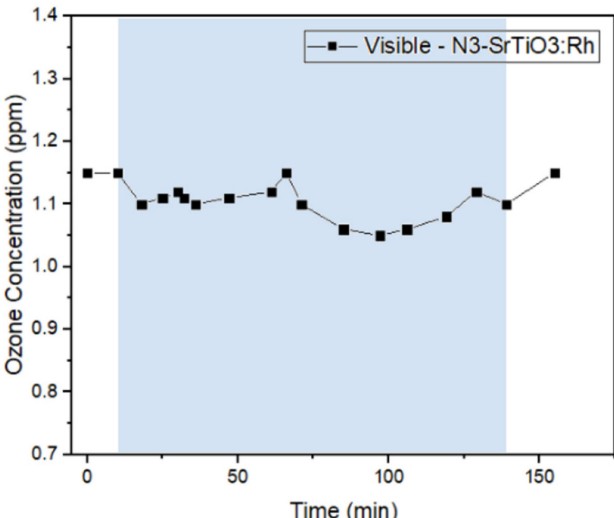

**Figure 5.** Photocatalytic decomposition of ozone in the continuous flow system by N3-SrTiO$_3$:Rh (N3 dye sensitized rhodium-doped strontium titanate) under the halogen lamp (OSRAM, 50 W, broad band, with 400–500 nm bandpass filter).

From Figure 3, there is an interesting phenomenon observed by using P25 under the visible light condition. In general, P25 can only absorb a little amount of visible light so it is expected to have poor ozone conversion. However, due to the large amount of hydroxyl groups on the surface of P25, when visible light is irradiated on the surface of P25, it will promote photoadsorption, which causes the ozone concentration decrease by enhanced adsorption on the surface [54–58]. Additionally, rutile crystallites among mixed phase P25 can lead to catalytic spots on the interface of rutile/anatase under visible light irradiation [59]. When the light remained on for a few hours, P25 reached the full adsorption capacity of ozone. Therefore, we can observe the concentration rise back when the light is still on, after that, the difference of ozone concentration between the final steady state value and the initial concentration is believed to be the contribution by photocatalysis. As for Figure 4, the ozone concentration only has a slight decrease when the light is turned on, which suggests that SrTiO$_3$:Rh has a mere photocatalytic activity on removing ozone under the visible light condition. This result may seem somehow contradictory to the results in the batch system. Yet, the ozone removal rate in the batch system shows the combined effect of adsorption and photocatalysis. On the other hand, in the continuous flow system, the adsorption/desorption will first reach a steady state before turning on the light, and the ozone concentration decrease will be totally caused by photocatalysis except the photoadsorption observed in Figure 3. Consequently, the result in Figure 4 can be reasonably explained that SrTiO$_3$:Rh has a relatively well ozone removal performance by the combination of adsorption and photocatalysis. However, the results in Figure 4 showed that it could still be further improved to have better ozone conversion. As a result, the N3 dye loaded SrTiO$_3$:Rh is prepared to enhance the photocatalytic ability of pure SrTiO$_3$:Rh. We expect that by immersing the photocatalyst into the dye so that N3 dye is physical adsorbed on the surface, it is able to increase the light absorption within the visible light region. In Figure 5, although there are some fluctuations on ozone concentration, N3-SrTiO$_3$:Rh can decrease the ozone concentration to an even lower extent.

The addition of the N3 dye further improved the photocatalytic reaction under the visible-light condition. Furthermore, since the reaction in this study occurred at the gas/solid interface, the degradation of the dye was found negligible under our experimental conditions.

A recent review arranged the existing literature of ozone decomposition [60]. Firstly, one technique using activated carbon based filters had high ozone conversion, i.e., >90%, but the filters were highly dependent on the activation carbon grade, the ozone concentra-

tion, the relative humidity and the residence time. Additionally, they would need to be replaced after a certain time of usage. Secondly, catalytic decomposition was the promising method to remove ozone. $MnO_x$ based catalysts and other noble metal based catalysts were reported to possess the ozone decomposition efficiency from <20% to 100% depending on the different reaction temperature, the relative humidity, the reaction time, and the ozone concentration. Most studies were conducted at a high ozone concentration, e.g., range from 20 to 20,000 ppm, which was far apart from the actual scenario. In addition, the reaction mechanism might be different due to the difference of the concentration gradient. Third, the photocatalytic decomposition was another technique introduced to remove ozone. $TiO_2$ based photocatalysts were the extensively studied material. The reported ozone decomposition efficiency ranged between <5% and >95% dependent on the structure of the material, gas flow rate, relative humidity, reaction temperature, etc. In our study, we proposed a novel idea on using a visible light photocatalyst on removing ozone under ambient condition. Furthermore, the testing concentration was controlled at a low concentration range, which was much similar to the real world situation. Although the conversion level is relatively low compared with the commercially available filters and catalysts, it has the advantages of the ability to function under room temperature, functions under visible light, and possess a certain amount of photoadsorption.

### 2.2.4. Conversion of Ozone in the Continuous Flow System

The conversion of ozone in the continuous flow system is calculated by applying the following Equation (1).

$$Ozone\ conversion = \frac{C_{dark} - C_{light}}{C_{dark}} \times 100\% \qquad (1)$$

where:

$C_{dark}$: initial ozone concentration when light is off (ppm);
$C_{light}$: the lowest ozone concentration when the light is on (ppm).

The calculated conversion results for continuous flow system under UV light and visible light were combined together in Table 2. From Table 2, under UV irradiation, P25 possessed 6.3% conversion of ozone, which is the highest conversion among the three catalysts used in continuous flow system UV tests. Under visible light irradiation, subtracting the photoadsorption removal of ozone of P25, both in-house sensitized $SrTiO_3$:Rh based catalysts possess higher conversion than that of commercial catalyst P25. Especially, N3 dye sensitized $SrTiO_3$:Rh has the highest conversion among the three catalysts used under visible light irradiation. To sum up, due to the lowering of the band gap by doping rhodium into the pristine $SrTiO_3$ structure, $SrTiO_3$:Rh could absorb visible light, and immersing $SrTiO_3$:Rh into N3 dye can enhance its visible light absorption by dye physical adsorption, which gives improved photocatalytic removal of ozone under visible light and ambient condition. Although most of the tests shown in the work were conducted under a high flow rate, 7250 L $g^{-1}$ $h^{-1}$, we recommend operating the system with a lower flow rate in order to optimize the performance of the ozone removal system.

### 3. Materials and Methods

#### 3.1. Catalyst Synthesis

The control group commercial catalyst was P25 (99.5%, Sigma-Aldrich, St. Louis, MO, USA). Additionally, the other two catalysts including $SrTiO_3$ and $SrTiO_3$:Rh were prepared in-house. The schematic of the procedure can be seen in Figure 6. The procedure was performed by using an advanced sol–gel method modified from the synthesize method proposed by Xuewen et al. [61]. First, mix the precursors including citric acid (99%, Sigma-Aldrich, St. Louis, MO, USA), strontium nitrate (99%, Showa Chemicals, Tokyo, Japan) and rhodium chloride (99.5%, Merck, Kenilworth, NJ, USA), which provides rhodium metal into the material structure. The number on the Figure 6 indicates the order of adding each

chemical. Next, add tetrabutyl titanate (TBOT, Alfa Aesar, 99%, Tewksbury, MA, USA) dropwise into the above mixture solution. Then, add a few drops of ethylene glycol to get a better dispersion of catalyst particles and adjust the pH value of the solution with nitric acid. Next, stir the solution for 25 h and dry it in an oven at 80 °C for 8 h. Last, pulverize the dried powder plate and calcine it at 1200 °C for 10 hours. As a result, the catalysts had been prepared by the modified sol–gel method to get the correct material structure and a smaller particle size with a higher surface area. Characterization of the $SrTiO_3$ based photocatalyst synthesized from several methods has previously been reported by Wu's group [62–64]. In addition, to increase visible light absorption of the catalyst, the dried $SrTiO_3$:Rh will be further immersed in the N3 dye solution. The main structure of N3 dye is shown in Figure 6. After immersing for 24 hours, the catalyst is then dried at 70 °C for six hours and obtained the N3-$SrTiO_3$:Rh (N3 dye sensitized $SrTiO_3$:Rh) with a shallow purple color.

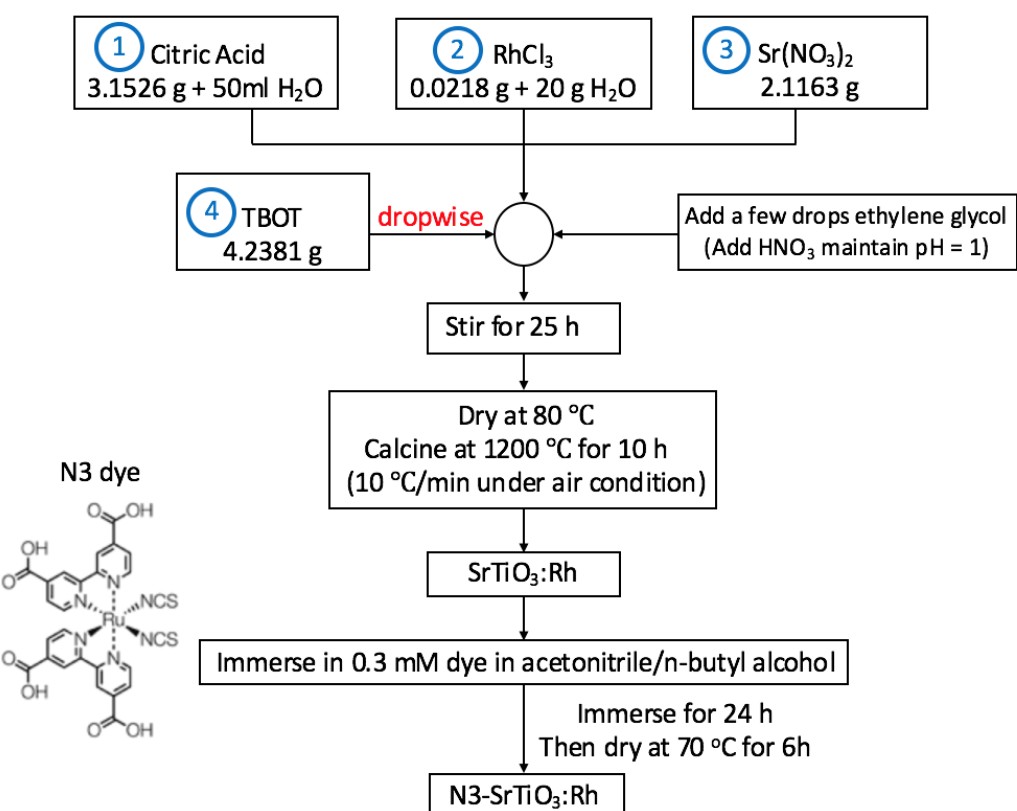

**Figure 6.** The catalyst synthesizes the procedure by using the modified sol–gel method.

### 3.2. Characterization

3.2.1. X-ray Diffraction (XRD)

The XRD (Rigaku, Tokyo, Japan) pattern (Figure 7) shows that each of these $SrTiO_3$ based materials had the same characteristic peaks, which indicates that all of them had a similar crystal structure [65]. However, due to the difference in calcination temperature and the solution pH value, the crystal sizes still possess a mere difference. In theory, the higher temperature results in a larger crystal size. In addition, the XRD pattern did not show any peaks indicating the doping of rhodium, which is because the doping amount of rhodium is too small to be observed by XRD.

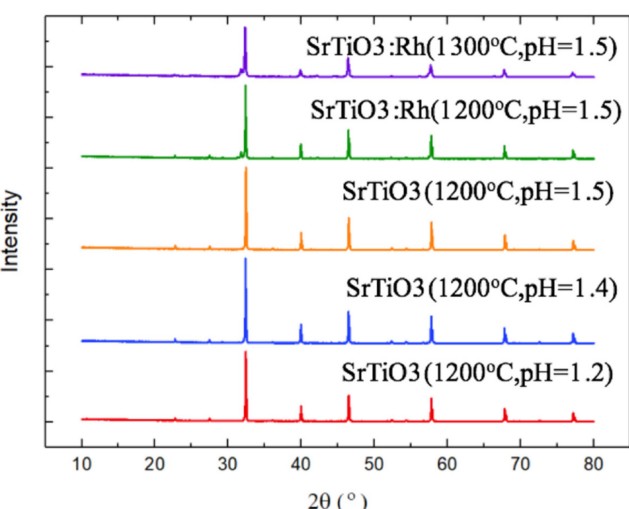

**Figure 7.** XRD results of the in-house synthesized catalysts. Different colors indicate different materials and different preparation conditions. The temperature indicates the calcination temperature of the materials and the pH value indicates the solution pH value during the sol–gel process.

### 3.2.2. UV–Vis Spectrometry

Our group synthesized several $SrTiO_3$ based catalysts under different parameters. However, in order to compare the photocatalytic ability of each catalyst under a similar synthesis environment, $SrTiO_3$ and $SrTiO_3$:Rh were mainly used to conduct the photocatalytic decomposition test. These two catalysts were both synthesized under 1200 °C calcination temperature and the pH 1.5 sol–gel solution environment.

According to Figure 8, the UV–Vis spectra (Agilent, Santa Clara, CA, USA) show an increased absorption of visible light in $SrTiO_3$:Rh comparing to $SrTiO_3$. Both materials have the ability to absorb lights in the UV region. For $SrTiO_3$:Rh, the spectrum indicates that rhodium has gone into the catalyst structure thus providing another donor level and reduces the bandgap of this material. Furthermore, the spectrum shows that $SrTiO_3$ could only absorb light with a wavelength shorter than 400 nm. This can also be inferred by the color change while $SrTiO3$ had a white color and it turned into gray when doped with rhodium. In comparison with another widely used photocatalyst $TiO_2$, either $TiO_2$ in the rutile form or anatase form only absorbs light in the UV region [66]. There are several other possible candidate materials, which demonstrate the ability of visible light photocatalysis, e.g., g-$C_3N_4$, $TiO_2$ doped with other elements [67,68].

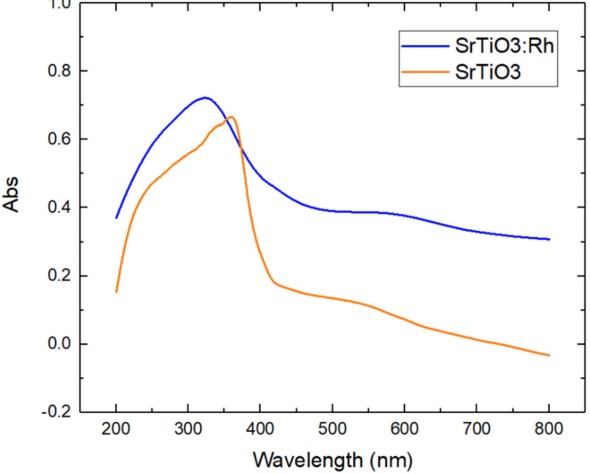

**Figure 8.** UV–Vis spectrum of strontium titanate (calcination temperature: 1200 °C, pH = 1.5) and rhodium-doped strontium titanate (calcination temperature: 1200 °C, pH = 1.5).

### 3.2.3. SEM/TEM and EDS Analysis

Figure 9 showed the morphology of photocatalyst, which was examined by the scanning electron microscope (SEM, JEOL JSM-7600F, Tokyo, Japan)and the transmission electron microscope (TEM, Hitachi H-7100, Tokyo, Japan). The particles are near round-cubic shape and the sizes are around 200–300 nm. The elemental analysis was measured by electron energy dispersive X-ray spectroscopy (EDS, JEOL JSM-7600F, Tokyo, Japan), which is shown in Figure 10 and listed in Table 3. The atomic ratios are near the stoichiometric of $SrTiO_3$. The Rh cannot be detected because of a very small amount of dopant.

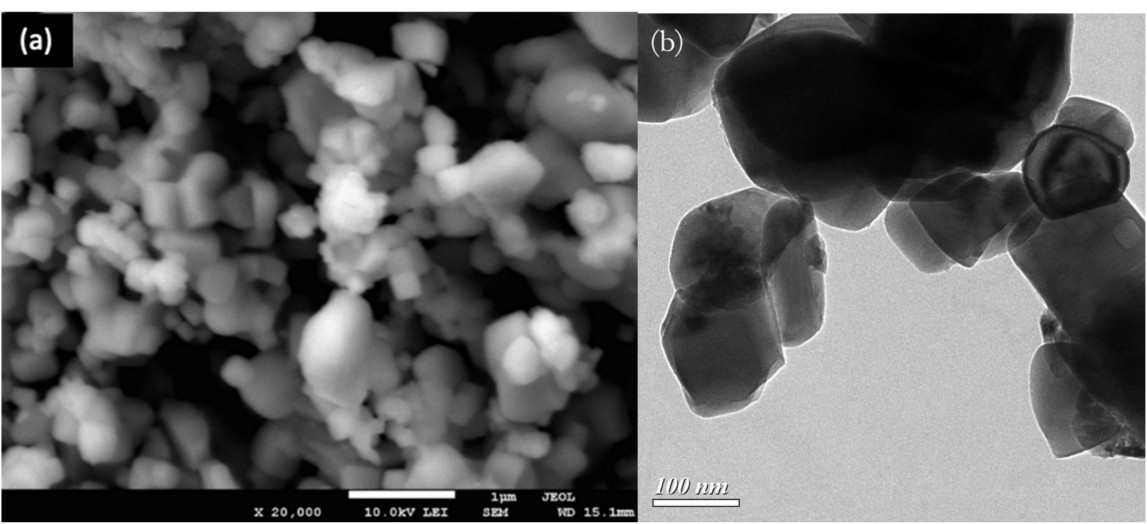

**Figure 9.** (**a**) SEM of $SrTiO_3$:Rh and (**b**) TEM of $SrTiO_3$:Rh.

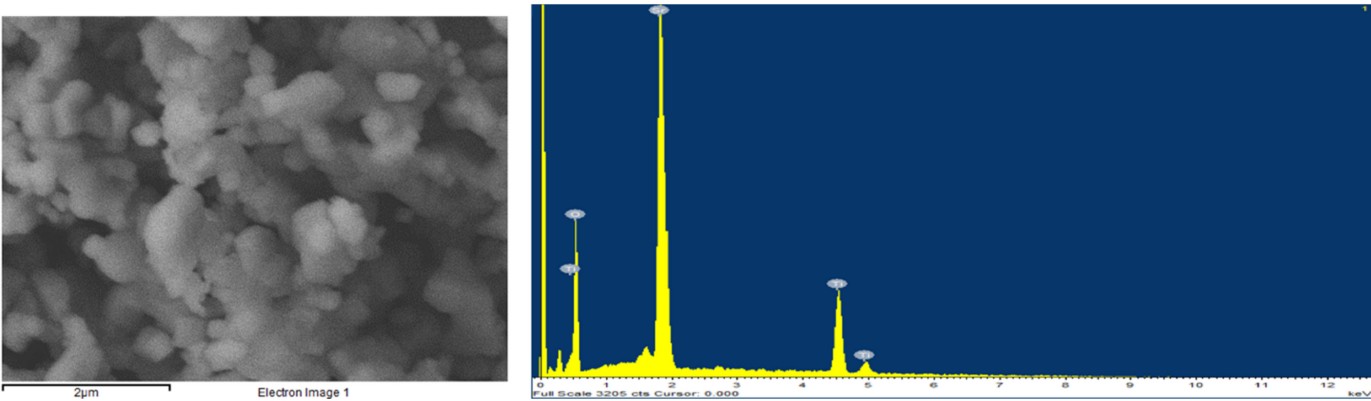

**Figure 10.** EDS analysis for $SrTiO_3$:Rh.

**Table 3.** Element composition ratios of $SrTiO_3$:Rh by EDS.

| Element | Weight (%) | Atomic (%) |
|---------|-----------|-----------|
| O | 24.6 | 57.8 |
| Ti | 27.5 | 21.6 |
| Sr | 47.9 | 20.6 |

### 3.2.4. BET Surface Area Analysis

The surface area of the $SrTiO_3$ catalyst was measured with the nitrogen adsorption method. The results showed that the catalyst possessed a surface area of 0.6598 $m^2$/g.

### 3.3. Experiment Setup

3.3.1. Batch System

The batch reactor system (Figure S1) contains the ozone producer (Ionkini, JO-6271, Guangzhou, China), ozone detector (Dräger, X-am 5000, Lübeck, Germany) and photo-catalysts. All of the above were put into the customized batch reactor before the ozone started to produce. The light source was connected to the reactor through an optical fiber, which can be illuminated from the top of the reactor. The experimental sequence was as follows. First, the ozone producer and detector were turned on, then they were carefully sealed in the batch reactor and started to monitor the ozone concentration rise. When the ozone concentration exceeded 0.12 ppm, the ozone producer was turned off and the ozone concentration decay profile was monitored by the ozone detector. An important factor is the ozone self-decay rate since it is a highly reactive chemical. The blank test indicated in each figure in Section 2 all contain the blank ozone decay, which shows the decay without adding photocatalysts. Besides the blank test, light irradiation was applied to the batch re-actor throughout the whole ozone photodecomposition experiment. The light source used in this experiment includes a fixed-wavelength (365 nm) UV light and a broad-band Xenon lamp. Three materials including $SrTiO_3$, $SrTiO_3$:Rh and P25 were tested in this experiment under ambient condition. The powder form photocatalyst was spread uniformly with the largest extent of area exposed to light and placed in the customized reactor.

However, the batch system has the defect that it is not similar to the continuous ozone production environment inside the factory. Furthermore, the distance between the detector and the producer will somehow influence the monitored concentration and cause the misevaluation of the ozone removal rate. In addition, it is unable to distinguish the effect of ozone removal between adsorption and photocatalysis in the batch system. After calculating the ozone removal rate and understanding the possible behaviors of the three materials used. The ozone photodecomposition system is further improved into the continuous flow system for a wider range of tests.

3.3.2. Continuous Flow System

In order to distinguish the effect of ozone removal between adsorption and photo-catalysis, the continuous flow system is applied to verify the photodecomposition of ozone under continuous production environment and to obtain the conversion of ozone.

As shown in Figure S2, the source of the ozone is air from a gas cylinder, it will first go through a customized chamber with an ozone producer to produce ozone by applying voltage to decompose the oxygen molecule to trigger ozone formation, a small fan is also settled in the chamber to act as an agitator to mix the gas well. Next, the gas flows downstream passing through a customized plug flow reactor with a catalyst filled in it. The light source is illuminated from the quartz window on top of the reactor. The gas will then pass through a rotameter to measure the system's flow rate. The concentration of the ozone is measured by the ozone electrochemical detector, and the hygrometer and thermometer are used to monitor the humidity and temperature of the reaction condition. The downstream of the system is also directed towards a beaker with water, which acts as a site of ozone discharge.

The concept of applying this system is to let the catalyst go through a fully adsorption process under dark condition and reach a steady state ozone concentration. Then turn on the light source and illuminate on the catalyst through a quartz window on the top of the reactor, which will therefore trigger the photocatalytic ability of the catalyst and decompose the ozone. After a certain time period, the light source will be turned off in order to check if the ozone concentration will rise back to the previous amount before light irradiation. Additionally, the above cycle will be applied several times to test the repeatability of each catalyst under ambient condition.

*3.4. Data Analysis*

Quantitative Analysis of Ozone Removal Rate

To quantify the ozone removal rate, the following Equation (2) is applied.

$$\frac{dm}{dt} = \frac{\rho V \left( C_i - C_f \right)}{m_{cat.}\Delta t} \tag{2}$$

where:

$C_f$: Final ozone concentration (ppm);
$C_i$: Initial ozone concentration (ppm);
$dm$: mass of ozone removal amount (mg);
$dt$: time required to remove all ozone in batch reactor (s);
$m_{cat.}$: catalyst amount used (g)
$V$: reactor volume (mL);
$\rho$: gas density (kg/m$^3$).

From Equation (2), the ozone removal rate can be calculated by monitoring the initial and final concentration, also the decay time should be recorded. In addition, with the reactor volume of 1670 mL and gas density 1.293 kg/m$^3$. The calculated ozone removal rate is regarded as the ozone removal amount per unit catalyst weight per unit time.

## 4. Conclusions

This research aimed to remove indoor ozone under ambient condition by the photocatalytic process. The ozone removal rate of strontium titanate based catalysts were quantified through the batch system and further compared to commercial TiO$_2$ P25. In the batch system under visible light irradiation, SrTiO$_3$:Rh possessed the highest ozone removal rate, which was nearly $9.3 \times 10^{-3}$ µmol·gcat$^{-1}$·min$^{-1}$. In addition, to distinguish between the effect of adsorption and photocatalysis, photoreaction was also conducted through continuous flow system. From the results of continuous flow system, focusing on the visible light irradiation results, commercial TiO$_2$ P25 had the lowest ozone conversion. Furthermore, SrTiO$_3$:Rh made in-house with the sol–gel method had a higher ozone conversion, and N3 dye sensitized SrTiO$_3$:Rh possesses the highest ozone conversion under visible light irradiation at ambient condition, which was caused by higher visible light absorption attributed to dye physically adsorbed on the surface of the photocatalyst. To the best of our knowledge, this is the first time for researchers to distinguish the effect of photocatalysis under visible light and physical adsorption in removing indoor ozone under ambient conditions. Furthermore, it also revealed that SrTiO$_3$:Rh based materials had the ability to remove indoor ozone by photocatalysis. The work to improve visible light ozone conversion is still ongoing.

**Supplementary Materials:** The following are available online at https://www.mdpi.com/2073-4344/11/3/383/s1, Figure S1: Schematic setup of the ozone photodecomposition batch system: (a) ozone producer, (b) ozone detector and (c) photocatalyst, Figure S2: Schematic diagram of the continuous flow system: (A) air cylinder, (B) ozone producer, (C) agitator, (D) plug flow reactor, (E) light source, (F) rotameter, (G) ozone detector, (H) hygrometer and thermometer and (I) ozone discharge beaker, Figure S3: Natural decay of ozone in the batch system (no catalyst, no light source), Figure S4: Photocatalytic decomposition of ozone in the batch system by SrTiO$_3$ under UV (365 nm) irradiation for three repeated tests, Figure S5: Photocatalytic decomposition of ozone in the batch system by SrTiO$_3$:Rh under UV (365 nm) irradiation for three repeated tests, Figure S6: Photocatalytic decomposition of ozone in the batch system by P25 under UV (365 nm) irradiation for three repeated tests, Figure S7: Photocatalytic decomposition of ozone in the batch system by SrTiO$_3$ under Xe lamp (500 W, broad band, with AM 1.5 G filter) irradiation for three repeated tests, Figure S8: Photocatalytic decomposition of ozone in the batch system by SrTiO$_3$:Rh under Xe lamp (500 W, broad band, with AM 1.5 G filter) irradiation for three repeated tests, Figure S9: Photocatalytic decomposition of ozone in batch system by P25 under Xe lamp (500 W, broad band, with AM 1.5 G filter) irradiation for three

repeated tests, Figure S10: Photocatalytic decomposition of ozone in the continuous flow system by P25 under UV lamp (365 nm), Figure S11: Photocatalytic decomposition of ozone in the continuous flow system by SrTiO$_3$ under UV lamp (365 nm), Figure S12: Photocatalytic decomposition of ozone in the continuous flow system by SrTiO$_3$:Rh under UV lamp (365 nm).

**Author Contributions:** Conceptualization, methodology, data curation, J.Q.S., Y.-C.C.; software, visualization, J.Q.S.; validation, formal analysis, investigation, resources, writing—original draft preparation, writing—review and editing, J.Q.S., Y.-C.C., J.C.S.W.; supervision, project administration, funding acquisition, J.C.S.W. All authors have read and agreed to the published version of the manuscript.

**Funding:** This research was funded by Taiwan MOST, grant number MOST 108-2221-E-002-111-MY3 and MOST 105-2221-E-002-206-MY3.

**Acknowledgments:** Authors thank the financial support from Taiwan MOST, grant number MOST 108-2221-E-002-111-MY3 and MOST 105-2221-E-002-206-MY3. Authors also show great appreciation to Mei-Rurng Tseng and Yu-Chieh Pao of Taiwan ITRI for providing the N3 dye and also Yi Shan Huang's assistance in synthesizing N3-SrTiO$_3$:Rh.

**Conflicts of Interest:** The authors declare no conflict of interest. The funders had no role in the design of the study; in the collection, analyses, or interpretation of data; in the writing of the manuscript, or in the decision to publish the results.

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
