# Peer review of "Visible-Light Photocatalyst to Remove Indoor Ozone under Ambient Condition"

_catalysts, doi:10.3390/catal11030383_

Round 1

Reviewer 1 Report

Visible Light Plasmonic Photocatalyst to Remove Indoor Ozone Under Ambient Condition

In this paper the authors have presented their research work on synthesis and characterization of novel visible-light photocatalyst to remove indoor ozone. Strontium titanate (SrTiO3), rhodium-doped strontium titanate (SrTiO3:Rh), and N3 dye treated SrTiO3:Rh were synthesized using modified sol-gel method and compared for their photocatalyst properties. The authors used both batch system and continuous flow systems to quantify the ozone removal rate and measure the ozone conversion amount, respectively. The authors showed that under visible light condition SrTiO3:Rh possess a highest ozone removal rate. In the continuous ozone flow system, the N3 dye sensitized SrTiO3:Rh catalysts possess the highest ozone conversion under visible light. The authors attributed this to the physical adsorption of dye on the surface of the photocatalyst.  The authors further state that these novel photocatalysts could replace current high-temperature thermal-catalysts for indoor applications in ambient conditions.

Comments:

  1. This is a well written paper with introduction, results, discussion and conclusions.
  2. Number of references is adequate, but references after 2018 are missing. Authors please add/replace newer references.
  3. Table 1. The variation in ozone conversion rate is rather high for the SrTiO3:Rh catalysts under visible light (although the authors are considering average values). Is there any reason behind this? Is this random error? What are the errors in these measurements?
  4. Page 6, lines 218-219: "....will be shown in Figure 3, S13, and S14." Do you mean "....will be shown in Figure 3, Figure 4, and Figure 5. "?
  1. The title has word Plasmonic in it. But the authors have not mentioned nor explained/used the word Plasmonic in entire paper. Why? What plasmonic properties of  SrTiO3:Rh has been used?
  1. Is it possible to dope SrTiO3 with different amounts of Rh and tune band gap of SrTiO3:Rh catalysts for further improvement in photocatalysis in visible light?
  2. Figure 8. The highest peak absorption for SrTiO3:Rh and SrTiO3 appear in the UV region and not in the visible region? Could you also add similar curve for TiO2 (both Rutile and Anatase) for benefit of the readers? Is it possible to use different base material (instead of SrTiO3) with absorption peak in the visible region?

Reviewer 2 Report

THe paper describes the synthesis and characterization of Sr-Rh/TiO2 photocatalysts, and their application for the remove of ozone under ambient conditions. THe results show that the Rh and Sr doped sample is able to successfully perform the ozone abatement in a continuous flow system with visible-light. Under my point of view, this is a nice study that deserves publication in this journal.

I only have a concern with respect the use of "plasmonic" in the title, since this is not discussed in the introduction section. 

Reviewer 3 Report

The manuscript by Quan Su et al. describes the synthesis and application of SrTiO3 based photocatalysts for batch and continuous flow remediation of O3. The authors explore the influence of doping with Rh and compare with P25 as reference. They also explore UV vs visible-light expanded illumination and the role of a dye as sensitizers.

The manuscript would benefit from a series of modifications/additions prior to final acceptance:

1) The title is misleading. No clear explanation about the influence of a "plasmonic" photocatalyst is introduced in this manuscript. An alternative title or better explanations should be included in the revised manuscript.

2) A better description of the experimental setup should be included in the manuscript. Additional images of the experimental setup would be very helpful.

3) The characterization of the catalysts is scarce. This aspect should be extended beyond XRD (that should be also expanded for better identification of phases). SEM, TEM, EDS should be included. How is the catalyst is deployed? Thickness? How is Rh distributed?

4) Section 2.1.4 should be better located in the Materials and Methods section.

5) A critical discussion regarding the obtained conversion values in comparison with existing literature, existing photocatalysts and existing alternative technologies should be included to enrich the manuscript.

6) How critical is the need of the dye? Can be degraded overtime? Please elaborate on this question and discuss if it is worthy in comparison with the use of P25.

7) Differences in comparison with the blank experiment (photolysis) are very narrow. Error bars should be included. 

Round 2

Reviewer 3 Report

The authors have improved different sections of the manuscript. There still are some missing characterization of the catalysts that should have been included in this manuscript instead of waiting for a forthcoming publication. If available, I would encourage the authors to complete the Supplementary Information with additional characterization techniques. The photocatalytic activity is very limited but given the overall comments of the other reviewers I am now in favor of accepting the manuscript for publication.

Author Response

See attached response to reviewer.
